# Early asymmetric cues triggering the dorsal/ventral gene regulatory network of the sea urchin embryo

Vincenzo Cavalieri*, Giovanni Spinelli*

Department of Biological, Chemical and Pharmaceutical Sciences and Technologies, University of Palermo, Palermo, Italy

**Abstract** Dorsal/ventral (DV) patterning of the sea urchin embryo relies on a ventrally-localized organizer expressing Nodal, a pivotal regulator of the DV gene regulatory network. However, the inceptive mechanisms imposing the symmetry-breaking are incompletely understood. In *Paracentrotus lividus*, the Hbox12 homeodomain-containing repressor is expressed by prospective dorsal cells, spatially facing and preceding the onset of *nodal* transcription. We report that Hbox12 misexpression provokes DV abnormalities, attenuating *nodal* and *nodal*-dependent transcription. Reciprocally, impairing *hbox12* function disrupts DV polarity by allowing ectopic expression of *nodal*. Clonal loss-of-function, inflicted by blastomere transplantation or gene-transfer assays, highlights that DV polarization requires Hbox12 action in dorsal cells. Remarkably, the localized knock-down of *nodal* restores DV polarity of embryos lacking *hbox12* function. Finally, we show that *hbox12* is a dorsal-specific negative modulator of the p38-MAPK activity, which is required for *nodal* expression. Altogether, our results suggest that Hbox12 function is essential for proper positioning of the DV organizer.

*For correspondence: vincenzo.
cavalieri@unipa.it (VC); giovanni.
spinelli@unipa.it (GS)

**Competing interests:** The authors declare that no competing interests exist.

**Reviewing editor**: Robb Krumlauf, Stowers Institute for Medical Research, United States

## Introduction

Patterning of the embryonic ectoderm along the dorsal/ventral (DV) axis, also known as oral/aboral axis, has been extensively studied in various species of sea urchins. DV polarity is not firmly established in the unfertilized egg, but rather relies on a combination of inherited maternal information and inductive interactions among early blastomeres, becoming morphologically recognizable from the gastrula stage onward (*Brandhorst and Klein, 2002*; *Angerer and Angerer, 2003*; *Molina et al., 2013*). The ectoderm of the pluteus larva is noticeably partitioned into four main domains: (1) the oral/ventral ectoderm, a thickened epithelium surrounding the mouth, (2) the aboral/dorsal ectoderm, a squamous epithelium that covers most of the rest of the larval body, (3) the ciliary band, a belt of ciliated cells positioned at the border between oral and aboral ectoderm, and (4) the apical neurogenic domain.

The genetic landmark of polarization along the secondary axis is the zygotic expression of the TGF-β superfamily member Nodal on the future oral side, which behaves as an organizing centre imposing DV polarity in all three germ layers of the embryo (*Duboc et al., 2004*; *Flowers et al., 2004*; *Duboc et al., 2010*; *Materna et al., 2013*). Targets of Nodal signaling within the oral ectoderm include genes encoding the TGF-β pathway extracellular components Lefty, BMP2/4 and Chordin (*Angerer et al., 2000*; *Duboc et al., 2004, 2008*; *Bradham et al., 2009*; *Lapraz et al., 2009*; *Yaguchi et al., 2010*). Although they are expressed by the same cells, Lefty is thought to diffuse more rapidly than Nodal, thus acting as a long-range Nodal inhibitor (*Bolouri and Davidson, 2010*; *Duboc et al., 2004, 2008*). The BMP2/4 ligand acts instead as a relay to specify the aboral ectoderm, to which its signaling activity is confined, due to inhibition of BMP2/4 reception by Chordin within the oral ectoderm (*Angerer et al., 2000*; *Duboc et al., 2004*; *Lapraz et al., 2009*; *Chen et al., 2011*).

**eLife digest** Embryos begin as a collection of identical cells. As the embryo develops further, the cells in different regions must take on different structures and roles in order to form the complex tissues and organs seen in the fully developed organism. Therefore, a key task in early development is to inform cells where they are in a developing embryo. Signaling proteins released by special groups of organizing cells are responsible for providing the information about where a cell is located. Networks of genes controlled by these proteins then inform embryonic cells of where they are and what they should, or should not, become.

One such signaling protein is called Nodal, and is needed to perform a number of tasks in the developing embryo, including helping to form the basic tissues of the organism. Many animals depend on Nodal to develop correctly—from mice and humans, to zebrafish and sea urchins.

During sea urchin development, Nodal establishes where the mouth of a larva forms, setting up what is called the dorsal/ventral axis of the embryo; this separates the front and back of the embryo. To do so, the Nodal protein is mostly produced at the front of the embryo. Although much is already known about the network of genes that the Nodal protein controls, the genes and proteins that ensure that the initial source of Nodal is present at the right time and place are largely unknown.

Another protein called Hbox12 was also thought to be important for setting up the dorsal/ventral axis. Now, Cavalieri and Spinelli reveal that Hbox12 regulates Nodal during the development of a sea urchin embryo. In the early developing sea urchin, the gene that produces Hbox12 is activated in the region of the embryo that will become its back, directly opposite where Nodal is present. This activation normally occurs just before the gene that produces Nodal is turned on. If the *hbox12* gene function is impaired, the Nodal protein is produced in both the front and the back sections of the embryo. Conversely, if Hbox12 is introduced into regions where Nodal is present, the amount of Nodal decreases. Furthermore, disrupting Hbox12 prevents any signs of the dorsal/ventral axis forming.

Cavalieri and Spinelli propose that Hbox12 inhibits the production of Nodal by briefly inactivating another protein that is required to activate the *nodal* gene. By doing so, Hbox12 sets up the dorsal/ventral axis by restricting Nodal to the cells that will make up the front half of the embryo.

Most complex organisms have asymmetric bodies, and failure to establish these body asymmetries can result in disease and other disorders in humans. Deciphering how the dorsal/ventral asymmetry in the sea urchin embryo is established should improve our understanding of how the mechanisms that form body shapes have evolved.

The amount of details available on molecular circuits that govern DV patterning downstream of *nodal* expression is growing rapidly (*Su, 2009*; *Saudemont et al., 2010*; *Yaguchi et al., 2010*; *Chen et al., 2011*; *Li et al., 2012, 2013*). As opposite, only fuzzy clues are known as to the early steps that trigger the DV gene regulatory network. According to the current models, at the early blastula stage a maternally related anisotropy in redox gradient would transiently inactivate the p38 kinase in the future dorsal ectoderm (*Bradham and McClay, 2006*; *Coffman, 2009*), somehow leading to activation of the maternal bZIP and Oct1/2 factors on the ventral side (*Nam et al., 2007*; *Range et al., 2007*; *Range and Lepage, 2011*). The *cis*-regulatory apparatus of *nodal* responds to these factors, as well as to the maternal positive inputs of SoxB1 and Univin (*Range et al., 2007*), directing the expression of the gene within a discrete sector of the ectoderm that corresponds to the presumptive oral ectoderm (*Duboc et al., 2004*; *Flowers et al., 2004*; *Saudemont et al., 2010*). Such a spatial expression profile is then consolidated by a positive feedback mechanism related to the Nodal signal transduction system, and by the concurrent Nodal-dependent production of the Nodal antagonist Lefty.

It should be noted that all of the known positive inputs converging on the *nodal cis*-regulatory apparatus are broadly distributed in the embryo as early as *nodal* transcription occurs, raising the question of whether additional negative regulators are involved in the initial repression of *nodal* in all but the oral territories. In strict accordance with this possibility, it has been shown that the transcription repressor FoxQ2 contributes, together with Lefty, to suppress *nodal* expression in the apical neurogenic ectoderm (*Yaguchi et al., 2008*). On the other hand, a similar negative function acting on *nodal* within the presumptive dorsal ectoderm cells has not yet been uncovered. An extremely interesting

candidate for this role is the zygotically-expressed *hbox12* homeobox-containing gene. In *Paracentrotus lividus*, *hbox12* expression precedes the onset of *nodal* transcription, then declines after the 60-cell stage and is no longer detectable by hatching (*Di Bernardo et al., 1995*; this paper). Intriguingly, *hbox12* transcripts are asymmetrically distributed along the DV axis, being confined in cells that become aboral ectoderm, as revealed by *cis*-regulatory analysis (*Cavalieri et al., 2008*). The observed pattern of expression is so far unique in sea urchin development, and suggests that the Hbox12 transcription factor could act as a precocious input within the gene regulatory network that directs DV patterning. In agreement with this hypothesis, we have previously shown that disrupting the function of the Otx activator, a driver of *hbox12*, downregulates *hbox12* transcription and dramatically affects embryo polarization along the DV axis (*Cavalieri et al., 2008*). Here we extend these findings providing more direct evidence indicating that *hbox12* is a key upstream gene in the symmetry-breaking sequence of events, functioning to prevent the ectopic activation of *nodal* transcription within the prospective dorsal side of the early sea urchin embryo.

## Results

### Ectopic expression of the hbox12 repressor disrupts morphogenesis and affects nodal transcription

The timing and spatial expression profile of *hbox12* described earlier (*Di Bernardo et al., 1995*) led us to initially hypothesize an involvement of such a gene in the negative control of *nodal* expression. This is further supported by directly comparing the dynamic temporal and spatial patterns of *hbox12* and *nodal* transcription during very early development. As expected, *hbox12* expression begins at least two cell divisions earlier with respect to that of *nodal* (*Figure 1A*). Double-labeling whole mount in situ hybridization (WMISH) at the early blastula stage showed that the spatial domains of expression of the two genes occupy opposite sectors of the embryo (*Figure 1B*). These results, coupled with those of the *cis*-regulatory analysis reported previously (*Cavalieri et al., 2008*), lead us to definitively conclude that *hbox12* is expressed in the prospective dorsal ectoderm of the early embryo. Thus, *hbox12* is expressed at the right time and in the right place to regulate *nodal* expression.

As a first approach to reveal the role of *hbox12* during embryogenesis, we microinjected into zygotes the synthetic full-length mRNA at dosage ranging from 0.01 to 0.4 pg. When eggs from the same batches were injected with equal amounts of the control out-of-frame *strim1* transcript (*Cavalieri et al., 2011*), or low amounts of the *hbox12* mRNA (<0.05 pg), the embryos developed normally (*Figure 2A–C*). By contrast, 50% (n > 1500) of the specimens ubiquitously expressing 0.1 pg of the functional *hbox12* mRNA exhibited a highly reproducible strong perturbation. Development of these embryos was apparently normal until the mesenchyme blastula stage (*Figure 2D*). At gastrula stage, when control embryos displayed a clear DV polarity as shown by the thickening of the ventral side and the symmetric ventral-lateral arrangement of the two primary mesenchyme cell (PMC) clusters (*Figure 2B*), embryos translating exogenous *hbox12* mRNA appeared quite rounded and their PMCs were irregularly dispersed into the blastocoel (*Figure 2E*).

A striking phenotype was even more apparent at the pluteus stage. Control-injected embryos were normal angular-shaped larvae exhibiting the characteristic bilateral symmetry (*Figure 2C*). By contrast, *hbox12*-injected embryos appeared almost spherical (*Figure 2F*). In these specimens, PMCs retained a certain biomineralizing activity, producing two calcareous elements which did not elongate as much as those of the control embryos at the same stage. These embryos displayed a straight archenteron and did not form a stomodeum. As judged by morphological observation, their ectoderm was composed of only a thick and a squamous epithelium, respectively coating the animal and the vegetal side of the larva.

Exactly the same phenotype was obtained following injection of similar amounts of a synthetic mRNA encoding for the chimeric repressor HD-En (*Figure 2G*), in which the homeodomain of Hbox12 was joined to the repressor domain of *Drosophila* Engrailed. It follows that Hbox12 normally function as a transcriptional repressor in the early embryo.

Altogether, the phenotypes showed in *Figure 2* are broadly similar to those obtained by the knock-down of *nodal* function (*Duboc et al., 2004*), again suggesting a potential negative effect of the Hbox12 transcription factor on *nodal* expression. Indeed, microinjection of equal amounts of either *hbox12* or *hd-En* mRNA caused a dose-dependent attenuation in the level of *nodal* transcript at a very similar extent, as revealed by qPCR analysis from *hbox12*-injected embryos at morula stage

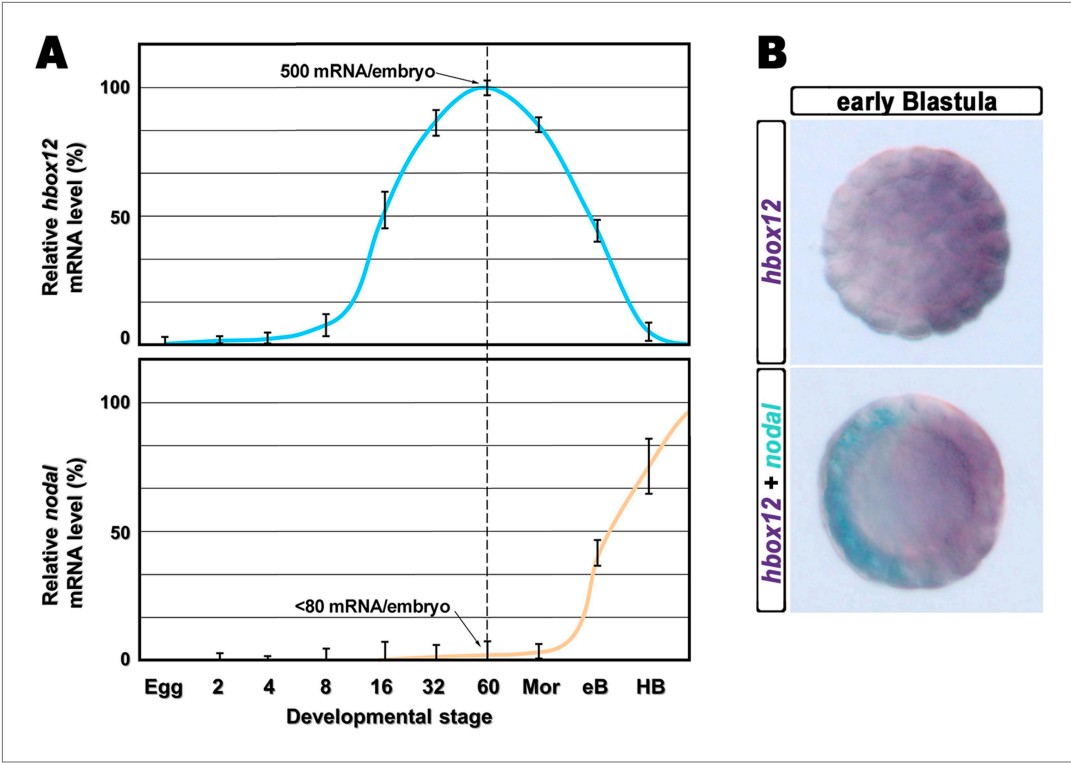

**Figure 1**. Expression of *hbox12* and *nodal* genes during early embryogenesis of *P. lividus*. (**A**) Temporal expression profiles examined by qPCR. Values at the different stages are shown as a percentage of the maximum signal intensity. Absolute numbers of transcripts per embryo given at the 60-cell stage are averages of the results of two independent experiments using distinct batches of cDNA. Abbreviations of the examined developmental stages: 2, 2-cell; 4, 4-cell; 8, 8-cell; 16, 16-cell; 32, 32-cell; 60, 60-cell; Mor, morula; eB, early blastula; HB, hatching blastula. (**B**) Spatial restriction of the *hbox12* and *nodal* transcripts observed following WMISH at the indicated stage.

(*Figure 2H* and *Figure 2—figure supplement 1*). Expression of the additional oral ectoderm marker *goosecoid* (*gsc*) as well as the aboral ectoderm marker *tbx2/3* was also reduced (*Figure 2I* and *Figure 2—figure supplement 1*), as expected from the perturbed expression of *nodal* normally necessary for the establishment of the entire DV axis (*Duboc et al., 2004*). The variations in transcript abundance shown in *Figure 2H,I* could appear somewhat modest, although they are of the same order of magnitude as those reported by other authors (*Agca et al., 2010*; *Bergeron et al., 2011*; *Coffman et al., 2004*). A possible explanation is that the qPCR data refer to the whole embryo population, affected and not affected. In fact, in our experiments only about half of the *hbox12*-injected embryos developed as aberrant larvae that, unfortunately, could not be isolated at early stages because they were virtually indistinguishable from the unaffected ones. In addition, the phenotype exhibited by the affected embryos was broadly similar but not perfectly identical to that obtained following knock down of *nodal* (*Duboc et al., 2004*), leading us to speculate that *nodal* expression was decreased to some extent in the overtly affected embryos. In close agreement with this hypothesis, WMISH showed that the expression of *nodal*, *gsc*, and *tbx2/3* genes was apparently reduced (n = 36/93; *Figure 2—figure supplement 2E–G*) or even nullified (n = 8/93; *Figure 2—figure supplement 2I–K*) in roughly 47% (n = 93) of *hbox12*-injected embryos. Although we were not able to quantify subtle differences, if any, in gene expression between control- and *hbox12*-injected embryos by a mere analysis of Dig-probe staining, the overall fraction of embryos showing defective expression of *nodal*, *gsc*, and *tbx2/3* was somewhat consistent with that of embryos that ultimately displayed unambiguous loss of DV polarization at later stages (*Figure 2E,F*). These findings substantially override the lack of significance in of qPCR expression data, that although showed less than threefold difference, are strengthened by the WMISH results. Altogether, our findings indicate that ectopic expression of *hbox12* disrupted the patterning of the ectodermal domains along the DV axis.

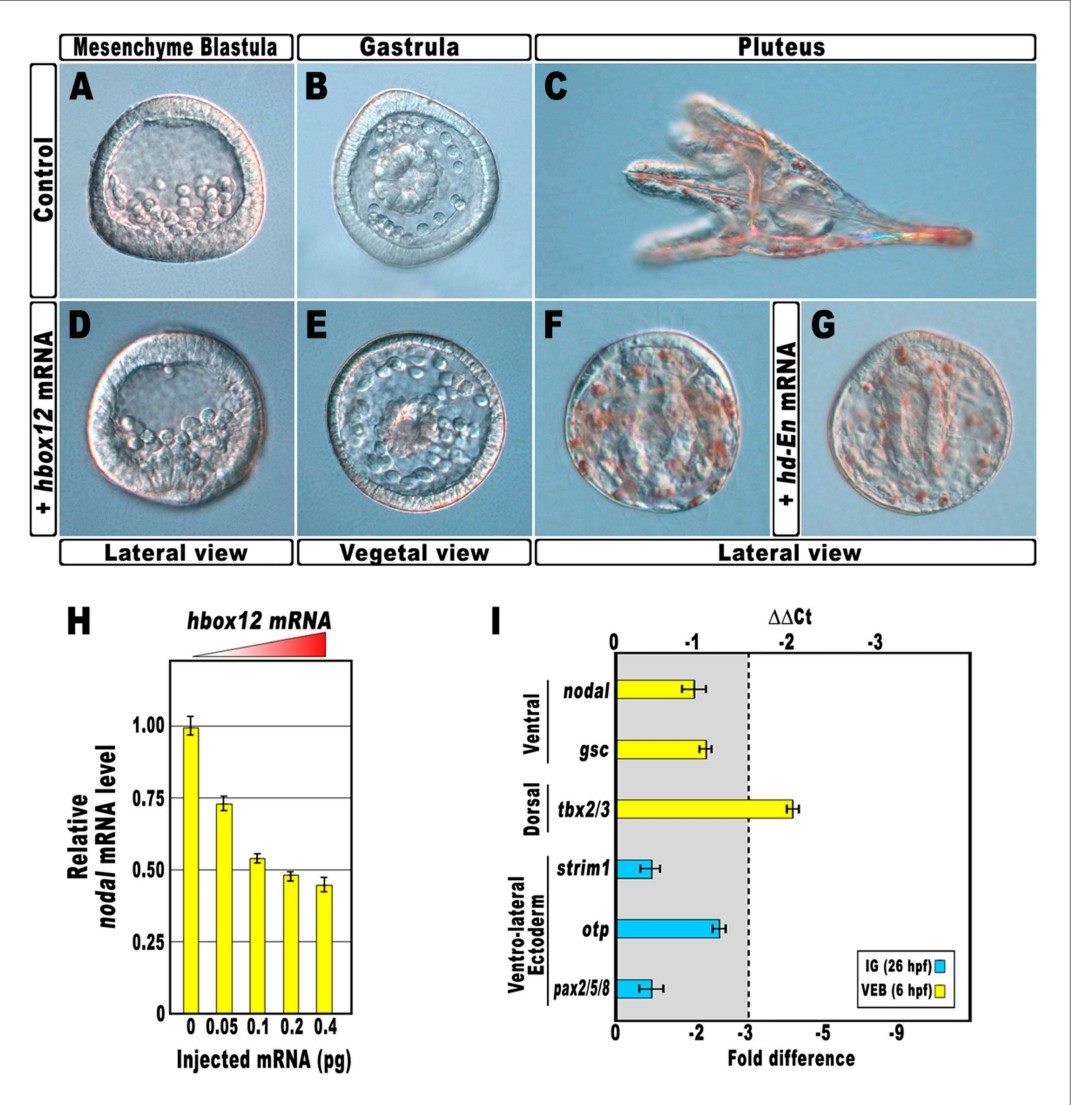

**Figure 2**. Disruption of embryonic DV polarity by ectopic expression of *hbox12*. (**A–G**) 0.1 pg of the full-length *hbox12* mRNA (**D–F**) or the *hd-En* mRNA (**G**), as well as a control out-of-frame *strim1* transcript (**A–C**), were injected into zygotes and embryos were observed at the indicated stages. Overexpression of either *hbox12* or *hd-En* severely perturbed DV axis formation, inflicting morphological defects that appeared from the gastrula stage onward. (**H**) qPCR measurements of *nodal* transcript abundance in embryos injected with increasing amounts of the *hbox12* mRNA. Values are shown as a percentage of the *nodal* mRNA level in control uninjected embryos. Further detail for the qPCR procedure is given in 'Materials and methods'. (**I**) Changes in gene expression level of *nodal* and other territorial marker genes assessed by qPCR in *hbox12*-injected embryos. Data are indicated as normalized ΔCt (ΔΔCt, left ordinate), and as the corresponding fold difference in transcript abundance (right ordinate), with respect to control embryos, at the same stage of development, derived from zygotes injected with the *strim1* out-of-frame transcript. The gray region represents ΔΔCt values corresponding to less than threefold difference. Although this is commonly considered the limit of significance for qPCR assays, the relevance of our measurements is reinforced by WMISH results (see text for details). Error bars are standard errors for the qPCR replicates. Oligonucleotide primer pairs used for qPCR reactions and amplicon lengths are indicated in *Supplementary file 1*. Abbreviations: lG, late gastrula; VEB, very early blastula.

The following figure supplements are available for figure 2:

**Figure supplement 1**. Overexpression of the HD-En obligate repressor and effect on *nodal* and *gsc* gene expression.

*Figure 2. Continued on next page*

*Figure 2. Continued*

**Figure supplement 2**. Spatial distribution of ectoderm- and PMC-specific markers in control and *hbox12* overexpressing embryos.

By WMISH with the PMC-specific marker *msp130* we also confirmed that in the *hbox12*-injected embryos the complement of PMCs was quite congruent with that of control embryos at the same stage but, in most embryos (70%, n = 250), the distribution of PMCs into the blastocoel was disorganized (compare panels J and K of *Figure 2—figure supplement 2*). This finding indicates that defects in PMC arrangement are due to a failure in the ectoderm to provide adequate patterning information. In close agreement, the transcript level of ectoderm-specific genes that have been previously implicated in skeletogenesis was reduced in *hbox12*-overexpressing embryos (*Figure 2I*).

## DV abnormalities arise from hbox12 misexpression in the animal hemisphere

As *hbox12* transcripts are present in a quadrant of the early embryo including the presumptive aboral ectoderm founder cells of the animal hemisphere and part of the veg1 tier (*Di Bernardo et al., 1995*; *Cavalieri et al., 2008*), we assessed *hbox12* misexpression in each domain by transplantation experiments. Zygotes were injected with the full-length *hbox12* mRNA together with the Texas Red-conjugated dextran (TRCD) lineage tracer and allowed to develop up to the 16-cell stage. Then, animal and vegetal halves from these embryos were separated and recombined with their complementary halves derived from control uninjected embryos (*Figure 3A*), and resulting chimeras were observed at pluteus stage. When *hbox12* was overexpressed exclusively in vegetal halves, almost all chimeras (n = 9/10) developed into larvae that were indistinguishable from controls (*Figure 3C*). In these embryos, descendants of vegetal cells normally formed the endomesoderm territories which, owing to TRCD, appeared red fluorescent.

By contrast, 6/10 of the reciprocal chimeric embryos, in which *hbox12* was present in the animal hemisphere (*Figure 3B*), produced the same phenotype as that observed with ubiquitous *hbox12* overexpression. Indeed, DV polarization was apparently impaired, stomodaeum did not form, and skeletal elements remained poorly elongated. Taken together, these findings reveal that overexpression of the transcriptional repressor Hbox12 in cells derived from the animal hemisphere impinges on the establishment of the DV axis.

## Establishment of the DV axis absolutely requires Hbox12 function

To study in more detail the role of *hbox12*, we first attempted to block the expression of the protein by injecting morpholino-substituted antisense oligonucleotides directed against the translation initiation site. Unfortunately, injected embryos always developed to the pluteus stage with no overt abnormalities. As multiple copies of the *hbox12* gene exist in *P. lividus* (*Cavalieri et al., 2008*), this failure may be due to gene copies with different translation initiation sites.

The Hbox12 protein includes the homeodomain close to the N-terminus and two serine-rich octapeptide repeats in the C-terminal region, which probably account for transcription repressor activity (*Cavalieri et al., 2008*). Therefore, to disrupt the *hbox12* function we expressed a truncated form of the protein, referred to as HD, which sequence ends just after the homeodomain. In principle, HD should efficiently compete with the endogenous Hbox12 for binding to DNA, quenching the repressor activity on target genes.

Again, almost all embryos injected with the control *strim1* transcript developed without deleterious effects (*Figure 4Aa–b*). By contrast, embryos injected with *hd* mRNA exhibited a failure of the DV axial patterning. At the gastrula stage, when controls exhibited a correctly partitioned ectoderm (*Figure 4Aa*), the vast majority of the *hd*-injected embryos (80%, n > 1200) appeared to be constituted by a uniformly thickened epithelium and no discernable ciliary band was identified (*Figure 4Ac*). The PMCs were homogeneously distributed around the straight archenteron, without distinguishable clusters, forming six to eight triradiate spicule rudiments (*Figure 4Ac–c'*). At the pluteus stage, HD-expressing embryos exhibited a range of phenotypes that could be ranked in order of severity. The most dramatically affected embryos (62%, n > 1200) have developed an archenteron. However, it never bent to fuse with the oral ectoderm but did grow from the centre of a spherical embryo (*Figure 4Ad*). In these specimens, PMCs adopted a full radial distribution, as confirmed by WMISH

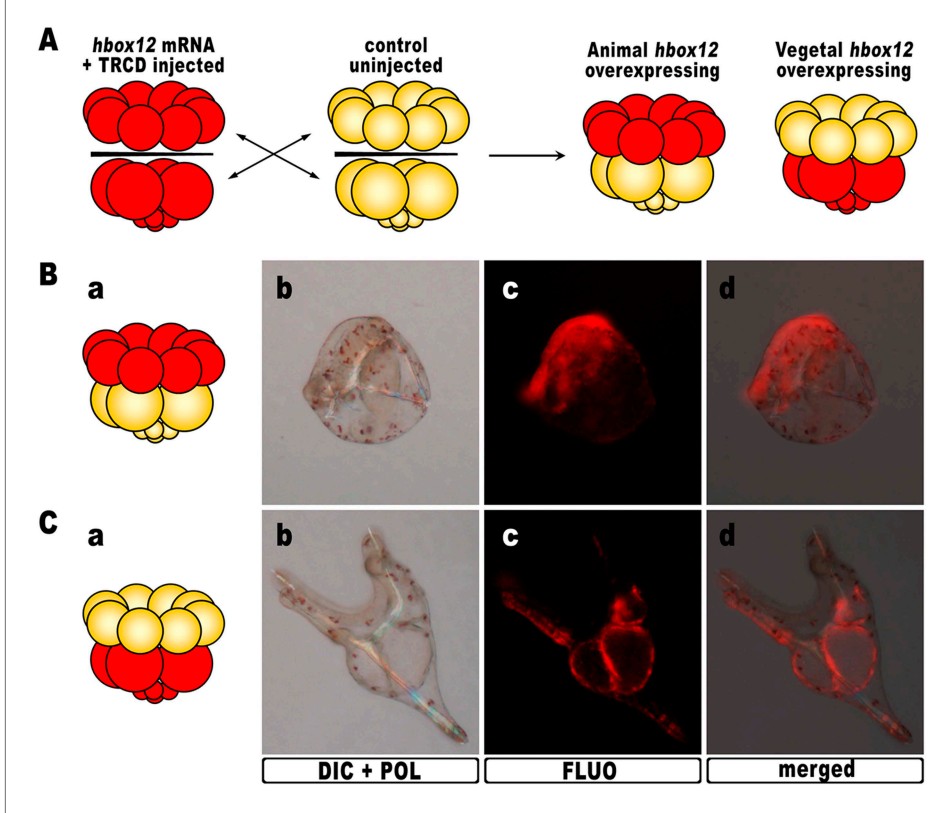

**Figure 3**. Overexpression of *hbox12* in chimeric embryos. (**A**) Schematic illustrating that, at the 16-cell stage, animal and vegetal halves of *hbox12*/TRCD-injected and control uninjected embryos were isolated and recombined. (**B–C**) Side views of representative examples of the resulting reciprocal chimeras examined at 48 hpf. The composition of the chimeras is shown in the diagrams in the left panels (**Ba** and **Ca**). Images for each embryo are shown under DIC optics with simultaneous plane polarized light illumination (**Bb** and **Cb**), epifluorescence (**Bc** and **Cc**), and aggregate merging (**Bd** and **Cd**).

with the *msp130* probe (***Figure 4Bh***, n = 31/45), and formed a grossly mispatterned skeleton around the circumference of the embryo (***Figure 4Ad–d'***). Moreover, pigment cells were not detected at all and embryos appeared uniformly albino (***Figure 4Ad***). A smaller fraction of *hd*-injected embryos (28%) showed a slightly less severe phenotype (***Figure 4Ae–e'***), indicating that axial specification was partially impaired in these embryos.

To ascertain the specification of different cell types in *hd*-injected embryos, we used a set of marker genes specific for the major embryonic territories. By WMISH, we first examined the expression of *nodal* and *gsc*, which are expressed exclusively in ventral ectoderm cells of control mesenchyme blastulae (***Figure 4Ba,4Bb***). In sharp contrast, we found that HD expression in embryos at the same stage caused broadened ectodermic transcription of both genes in approximately 70% (n > 80) of the resulting embryos (***Figure 4Be,4Bf***). The mRNA abundance of *nodal* and *gsc* was examined by qPCR and, as expected, upregulation was detected for both genes during development (***Figure 4C***).

The precise mechanisms that dictate the physiological function and target range of individual homeodomain proteins are in general either unknown or incompletely delineated (***Svingen and Tonissen, 2006***). As the DNA-binding properties of the homeodomains could not be, by themselves, sufficiently discriminating to distinguish between different sets of target genes in vivo (***Hoey and Levine, 1988***), to assess the specificity of HD on *nodal* and *gsc* gene transcription, we injected an equal amount of *otp-hd*, a synthetic mRNA coding for the closely related homeodomain of Orthopedia (***Di Bernardo et al., 1999***; ***Cavalieri et al., 2003***). Otp and Hbox12 homeodomains belong both to the Q50 class and show a very similar helix-III (***Figure 4—figure supplement 1***). As shown in ***Figure 4—figure supplement 2***, all the early developing embryos expressing such an isolated DNA-binding domain did not

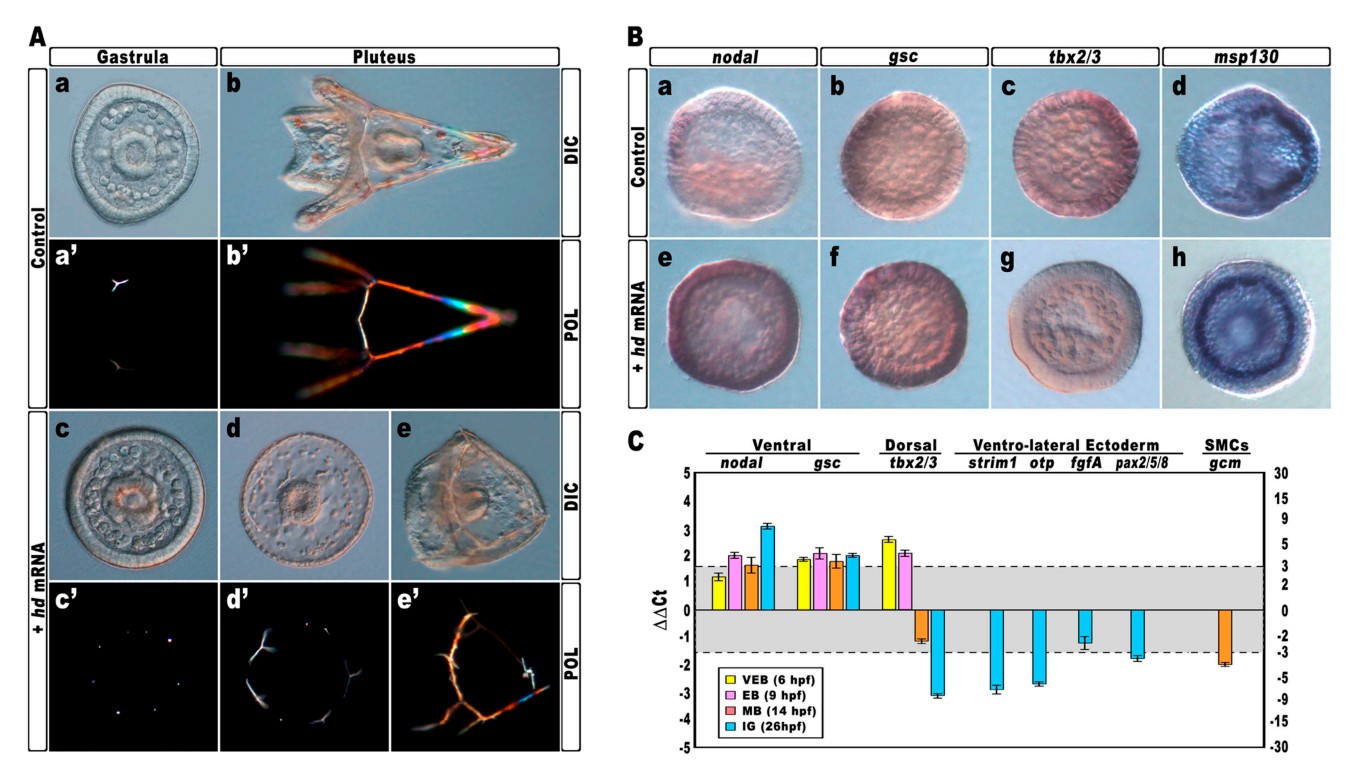

**Figure 4**. Impairing *hbox12* function and effects on DV axis formation. (**A**) Zygotes were injected with the control out-of-frame *strim1* RNA (**Aa–Ab**) or the *hd* mRNA (**Ac–Ae**), and the resulting embryos were observed from a vegetal view at the indicated stages. (**B**) Control- (**Ba–Bd**) and *hd*-injected (**Be–Bh**) embryos were fixed at the mesenchyme blastula stage and analysed by WMISH with the indicated probes. The embryo shown in (**Ba**) is oriented in a lateral view, while all the other embryos are in a vegetal view. (**C**) Changes in gene expression level of territorial marker genes assessed by qPCR during development of *hd*-injected embryos. Data are normalized and indicated as in *Figure 2I*. The gray region represents ΔΔCt values corresponding to non-significant variation (less than threefold difference). See also *Supplementary file 1*. Abbreviations: VEB, very early blastula; EB, early blastula; MB, mesenchyme blastula; lG, late gastrula.

The following figure supplements are available for figure 4:

**Figure supplement 1**. Multiple comparison of the homeodomain of Hbox12 and Otp proteins from *P. lividus*.

**Figure supplement 2**. Overexpression of isolated homeodomains and effect on *nodal* and *gsc* gene transcription.

display perceptible phenotypic aberration with respect to control injected embryos at the same stage. Most important, the transcript abundance of both *nodal* and *gsc* did not change significantly in embryos expressing the *otp-hd*. Altogether, these findings imply that impairing Hbox12 function through expression of HD specifically biased DV patterning toward ventralization.

We also noted that the mRNA abundance of the dorsal specific gene *tbx2/3* did fluctuate following a peculiar trend during development of *hd*-injected embryos. Although it was dramatically increased at early blastula stages (*Figure 4C*), it appeared significantly down-regulated at mesenchyme blastula stage (*Figure 4Bg*, n = 34/42, and *Figure 4C*) and almost completely abolished at late gastrula stage (*Figure 4C*). It should be emphasized that *tbx2/3* is expressed at a very low level during early embryogenesis (*Chen et al., 2011*; *Ben-Tabou de-Leon et al., 2013*). Hence, the simplest interpretation of the qPCR results is that the initial rise in *tbx2/3* transcription level could be due to an early pulse of BMP2/4 before the HD-induced overexpression of Nodal swamps the system. In close agreement, it has been shown in *P. lividus* that injection of *nodal* mRNA abrogates the expression of *tbx2/3* across the gastrula-stage embryo (*Duboc et al., 2004*, *2010*; *Saudemont et al., 2010*).

Consistently with the derepression of *nodal*, the expression of four ventro-lateral ectoderm specific markers, *strim1*, *fgf*, *otp* and *pax2/5/8*, was robustly down-regulated in *hd*-injected gastrulae

(*Figure 4C*), strongly supporting the hypothesis that ectoderm patterning was impaired in these embryos.

As described, an adjunctive defect characterizing most of the *hd*-injected embryos consisted in the lack of pigment cells. Known to be a cohort of secondary mesenchyme cells which expresses the specific marker *gcm*, the pigment cell precursors occupy a dorsal sector of the vegetal plate at the late mesenchyme blastula stage (*Ransick et al., 2002*; *Röttinger et al., 2006*; *Duboc et al., 2010*). At this stage, *hd*-injected embryos had downregulated *gcm* expression (*Figure 4C*). It is known that Nodal signaling antagonizes the specification of pigment cells (*Duboc et al., 2010*), and our result is perfectly congruent with the reported loss of *gcm* expression, and the albino phenotype, provoked by the ectopic expression of *nodal* across the embryo (*Duboc et al., 2010*). On this basis, we plausibly assume that the HD-induced overexpression of *nodal* similarly affected the specification of pigment cells in HD-expressing embryos.

## Hbox12 function is necessary in presumptive aboral ectoderm cells to ensure DV polarization

*Hd* transcript was injected into zygotes along with the TRCD-fluorescent tracer and then, at the 16-cell stage, animal and vegetal halves from these embryos were separated and recombined with their complementary halves derived from uninjected embryos. When HD was expressed only in vegetal halves, the resulting chimeras developed into normal plutei (n = 10/10), which endomesodermal territories appeared red-fluorescent (*Figure 5B*). Thus, *hbox12* function is not required in the vegetal half for DV polarization. By contrast, the reciprocal chimeric embryos, in which *hbox12* function was disrupted in the animal hemisphere, phenocopied the morphologies of the non-chimeric *hd*-injected embryos, and the strongly perturbed phenotype prevailed (n = 7/10; *Figure 5A*). These results support the contention that inhibition of *hbox12* activity in the animal hemisphere, where it is normally expressed, is sufficient to impede the establishment of the DV axis. In addition, these findings indicate that skeletogenic disorders originate not in PMCs, but are due to a failure in the overlying ectoderm to provide adequate patterning information.

To further strengthen confidence in this evidence, we attempted to disrupt the *hbox12* activity exclusively in aboral ectoderm founder cells. This was accomplished by introducing into zygotes the expression construct *phbox12*-HD-GFP, in which the HD-GFP fusion protein was specifically placed under the control of the wild type *cis*-regulatory sequences of the *hbox12* gene. This construct or *phbox12*-GFP, the control lacking HD (*Cavalieri et al., 2008*; see also 'Materials and methods'), was injected into zygotes and developing embryos were scored for GFP expression. In this experiment, the transgenes were delivered along with TRCD, to discriminate among injected embryos and those that escaped microinjection (not-fluorescent).

As expected, both constructs were expressed at a similar extent during early embryogenesis, as indicated by roughly comparable green fluorescence of embryos observed at the early blastula stage (*Figure 6—figure supplement 1A–B*). In agreement with previous observations (*Cavalieri et al., 2008*), at the gastrula stage green fluorescence was specifically detected in the ectoderm of an average of 60% (n > 500) of injected embryos (*Figure 6A–B*). However, whereas *phbox12*-GFP expression occurred in large ectoderm patches (*Figure 6Ac–d*), scattered and less fluorescent cells were observed in *phbox12*-HD-GFP-injected embryos (*Figure 6Bc–d*). This evidence and the fact that the number of HD-GFP-stained cells progressively extinguished as development proceeded (not shown) most likely indicate a low stability of the chimeric protein. Strikingly, despite this circumstance, most of the *phbox12*-HD-GFP-injected gastrulae expressing the transgene did show an equally thickened ectoderm and synthesized supernumerary spicule rudiments (*Figure 6Ba–b*), resembling embryos at the same stage that received exogenous *hd* mRNA (*Figs 3Ac–c′ and 4Ab–c*).

To obtain statistically relevant results we scored thousands of injected embryos at the early blastula stage. Specimens expressing HD-GFP at high level were transferred into a distinct plate filled with filtered sea water, and finally scored for the phenotype at 48 hr post-fertilization (hpf). At this stage, all the *phbox12*-GFP injected embryos were normal pluteus larvae expressing the transgene reporter in their aboral ectoderm (n > 500; *Figure 6C*). By contrast, about one half (n > 800) of the *phbox12*-HD-GFP-injected embryos phenocopied the abnormalities observed following *hd* mRNA overexpression (*Figure 6D,E*).

Despite the expected absence of GFP fluorescence at this stage (*Figure 6Dd,6Ed*), most of the affected embryos (78%) displayed profound defects in the establishment of the DV polarity, as shown

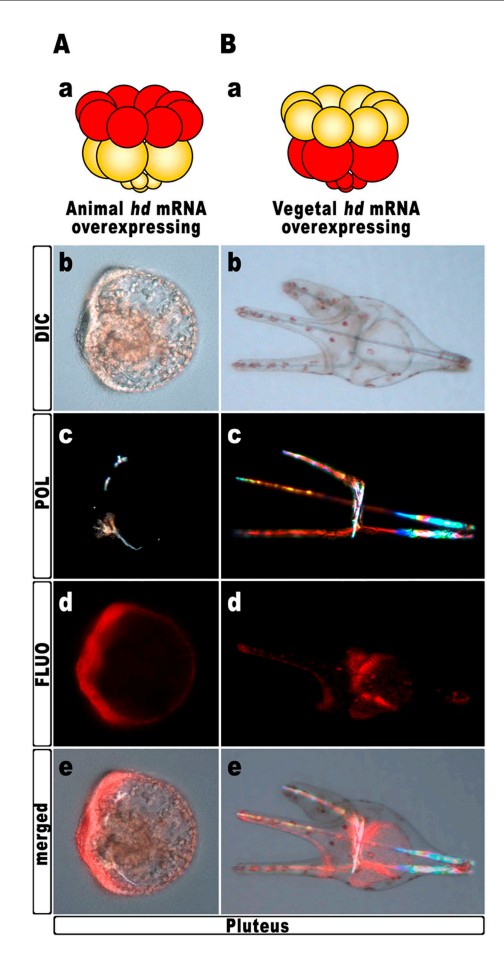

**Figure 5**. Block of *hbox12* function in the animal hemisphere of chimeric embryos, and effects on DV patterning. (**A–B**) Diagrams in (**Aa**) and (**Ba**) show the composition of the reciprocal chimeras resulting from animal and vegetal half recombination of *hd*/TRCD-injected and control uninjected embryos at the 16-cell stage. (**Ab–Ae** and **Bb–Be**) Side views of the resulting chimeras examined at 48 hpf. Images for each embryo are shown under DIC optics (**Ab** and **Bb**), plane polarized light illumination (**Ac** and **Bc**), epifluorescence (**Ad** and **Bd**), and aggregate merging (**Ae** and **Be**).

by the absence of the larval arms and aboral vertex, and the reduced number of pigment cells (*Figure 6E*). Worth mentioning, more than 60% of these embryos maintained such an abnormal phenotype for up to a week when maintained in culture, excluding that they were stunted embryos that could become normal with more developmental time. Otherwise, a smaller fraction (22%) of the *phbox12*-HD-GFP-injected embryos showed rather severe radialization coupled to the presence of multiple ectopic spicules (*Figure 6D*).

In this experimental assay, the prevalence of the milder phenotype could be explained by the combination between the mosaic incorporation of the exogenous DNA construct in the embryo (*Flytzanis et al., 1985*; *Franks et al., 1988*; *Hough-Evans et al., 1988*), and the rapid turnover of the HD-GFP protein.

The number of the dark-red pigment cells was quantified in late stage pluteus larvae using DIC optics. The total average number of these cells was about 45 ± 5 in *phbox12*-GFP-expressing larvae developed from two distinct batches of zygotes (n = 35 embryos counted in each experiment). This number did not differ from that observed in control uninjected plutei. In striking contrast, pigment cell population was greatly decreased in more than one half of *phbox12*-HD-GFP injected embryos observed at the same stage. In particular, specimens showing the milder phenotype differentiated less than 16 pigment cells (*Figure 6E*), and this number did not vary culturing the embryos up to a week (not shown). No pigment cells were instead detected in the *phbox12*-HD-GFP injected embryos exhibiting the fully radialized phenotype (*Figure 6D*), once again supporting the hypothesis that loss of *hbox12* function biased axial specification toward ventralization.

## The localized knock-down of nodal restores DV polarization in HD-expressing embryos

The results described in the previous sections strongly suggest that *hbox12* acts upstream of *nodal*, being involved in the asymmetrical establishment of the DV organizing centre. To better demonstrate the specificity of such a functional relationship, we performed a rescue assay in which the spatially restricted knock-down of *nodal* was superimposed on HD-expressing embryos. The experimental assay is depicted in *Figure 7A*. The *hd-GFP* mRNA was first microinjected into zygotes. At the 4-cell stage, the morpholino-oligonucleotide directed against *nodal* mRNA sequence (Mo-*nodal*) (*Duboc et al., 2004*) was successively injected into a single randomly chosen blastomere. To follow the fate of the re-injected cells, Mo-*nodal* was delivered together with the rhodamine-labelled dextran (RLDX) tracer. As expected, embryos injected with the *hd-GFP* mRNA and observed at the hatching blastula stage showed a diffused green fluorescence, indicating that the synthetic transcript was efficiently translated throughout the early embryo (*Figure 6—figure supplement 1C*). Since, as mentioned, the abundance of the HD-GFP protein is rather quickly eroded following embryogenesis, early

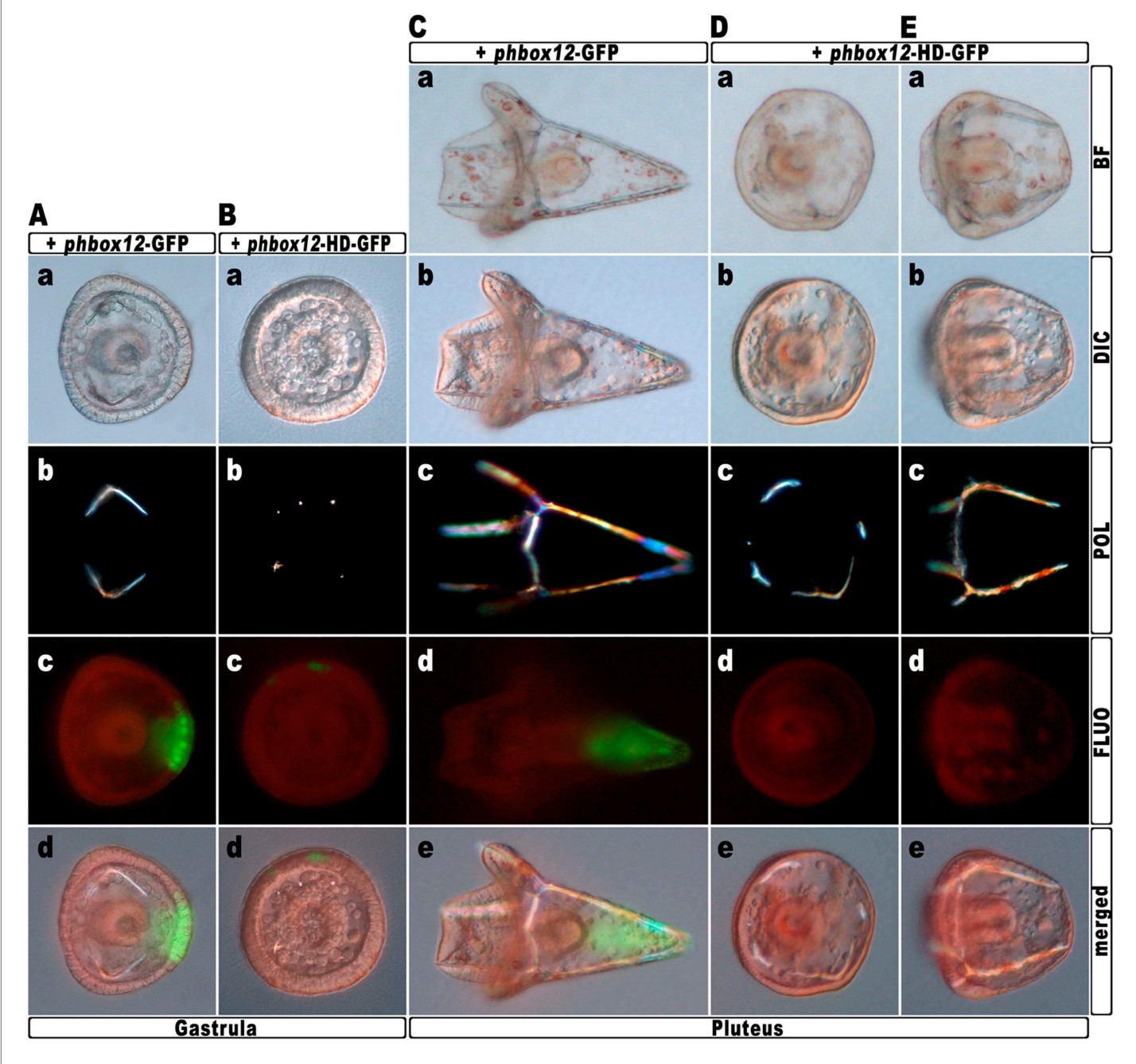

**Figure 6**. Block of *hbox12* function in aboral ectoderm cells of transgenic embryos, and effects on DV patterning. (**A–E**) Zygotes were injected with the indicated transgenes, and the resulting embryos were observed from a vegetal view at the indicated stages. Bright field (**Ca**, **Da**, and **Ea**), DIC (**Aa**, **Ba**, **Cb**, **Db**, and **Eb**), Dark field (**Ab**, **Bb**, **Cc**, **Dc**, and **Ec**), epifluorescence (**Ac**, **Bc**, **Cd**, **Dd**, and **Ed**), and aggregate merging (**Ad**, **Bd**, **Ce**, **De**, and **Ee**) images are shown.

The following figure supplement is available for figure 6:

**Figure supplement 1**. Expression of the HD-GFP fusion protein during early embryogenesis.

developing specimens simultaneously displaying GFP and RLDX fluorescence were selected for microscopic observation.

Sister batches of zygotes were injected with either the *hd-GFP* mRNA or Mo-*nodal* alone, and observed at 48 hpf. Once again, the former embryos exhibited a typically ventralized phenotype

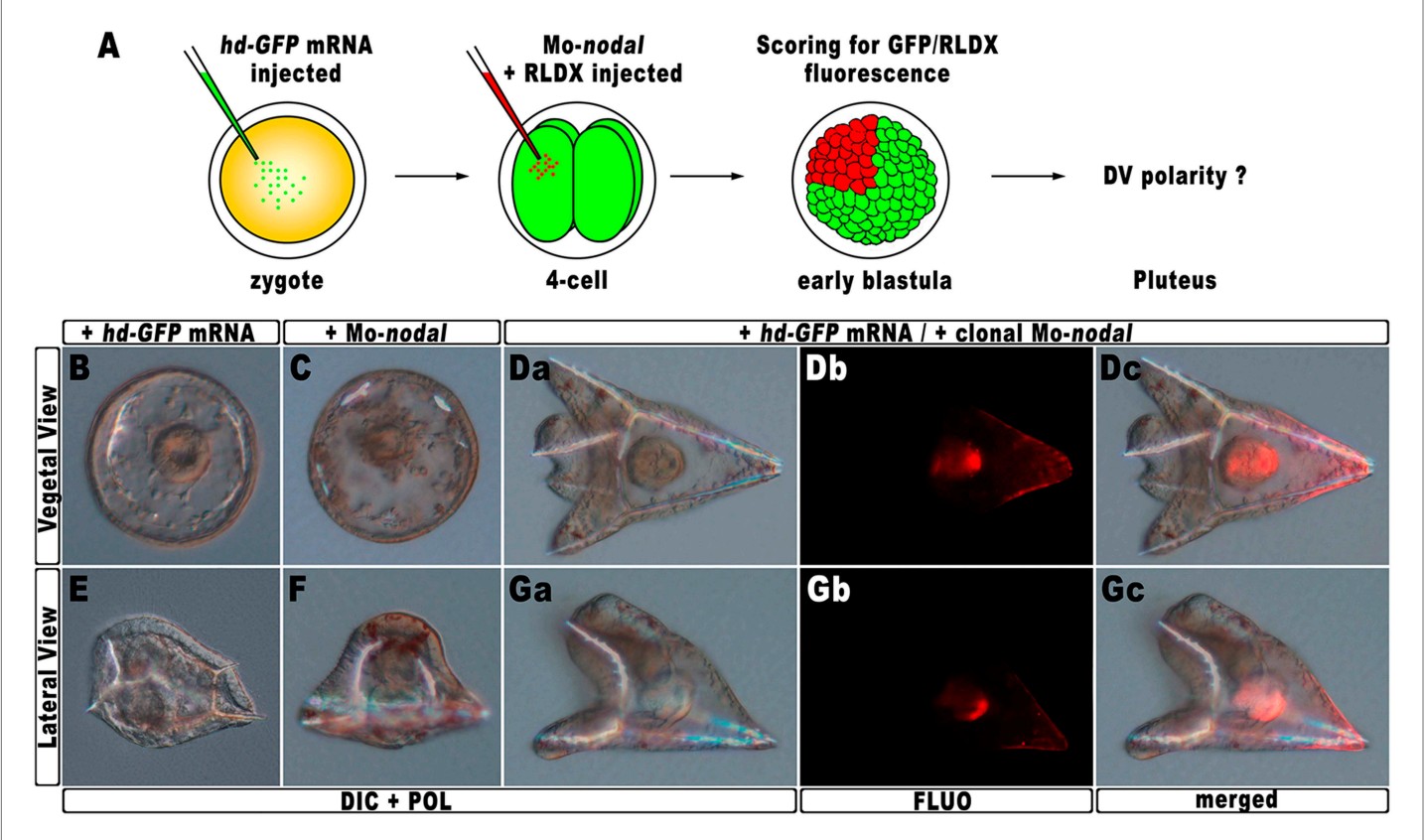

**Figure 7**. Rescue of DV polarity by clonal knock-down of *nodal* into *hd*-injected embryos. (**A**) At the 4-cell stage, one blastomere of *hd-GFP* mRNA injected embryos was co-injected with a morpholino oligonucleotide directed against *nodal* (Mo-*nodal*) together with the RLDX red fluorescent tracer. The resulting embryos were scored for simultaneous GFP/RLDX fluorescence at the early blastula stage, and eventually examined by microscopic observation at the pluteus stage. (**B–G**) Representative examples of embryos injected with either the *hd-GFP* mRNA (**B** and **E**) or the Mo-*nodal* alone (**C** and **F**), and of double-injected rescued embryos (**D–G**). Note that in both the rescued pluteus larvae shown in (**D**) and (**G**), the progeny of the blastomere that received Mo-*nodal* was embedded into the dorsal structures.

(*Figure 7B,E*), while embryos resulting from Mo-*nodal* injection developed into bell-shaped larvae with multiple entangled spicules (*Figure 7C,F*). In these embryos, most of the ventral and dorsal ecto-derm was replaced by a thick ciliated epithelium which, as demonstrated by other authors (*Duboc et al., 2004*), represents the default state of the ectoderm. Thus, both *hd-GFP* and Mo-*nodal* injected embryos, albeit with peculiar differences, never acquired any bilateral symmetry. Conversely, almost all double-injected embryos (n > 250) developed into normal pluteus larvae with a harmoniously patterned DV axis (*Figure 7D,G*). Remarkably, inspection of these embryos under fluorescence illumi-nation clearly revealed that the progeny of the blastomere injected with Mo-*nodal* was always found on the dorsal face of the rescued pluteus larvae (*Figure 7Db–c,7Gb–c*).

As it has been extensively shown that injection of Mo-*nodal* in *P. lividus* embryos abrogates trans-lation of the Nodal ligand (*Duboc et al., 2004*, *2010*; *Range et al., 2007*; *Saudemont et al., 2010*; *Bessodes et al., 2012*), we reasonably infer that in the double *hd*-GFP and Mo-*nodal* injected embryos, clonal knock-down of *nodal* was able to restore the asymmetrical production of the Nodal ligand, which in turn sufficed to resume the entire DV axis.

## Hbox12 expression negatively regulates the activity of the p38 MAPK

According to current models, the activation of *nodal* requires a phosphorylation event catalyzed by the p38 MAPK (*Bradham and McClay, 2006*; *Nam et al., 2007*; *Range et al., 2007*; *Coffman et al., 2009*). As mentioned, p38 is globally active, but is transiently inactivated in the prospective dorsal side of the early embryo, immediately before the onset of *nodal* expression (*Bradham and McClay, 2006*).

By this time, that in *P. lividus* corresponds to the 60-cell stage, the peak of *hbox12* transcription has just been accomplished (*Figure 1*), and therefore, anisotropic inactivation of p38 should correlate with *hbox12* expression. To confirm that prediction, the impact of perturbing *hbox12* function on p38 activity was determined by expressing a GFP-tagged p38, to reflect kinase activation as GFP nuclearization in living embryos (the inactive kinase is instead cytoplasmic).

Embryos expressing such a fusion protein developed normally whereas, at the 60-cell stage, p38 was cleared for a brief interval from nuclei on one side of the embryo (*Figure 8A*), which is thought to be the dorsal side. Impairing Hbox12 function, by co-injection of the *hd* mRNA, efficiently supplanted the p38 asymmetry, as shown by the uniform nuclear GFP staining in roughly 80% (n > 250) of the resulting embryos observed at the 60-cell stage (*Figure 8B*). Thus, loss of *hbox12* function revoked the transitory phase of p38 inactivation in dorsal cells. Reciprocally, uniform ectopic expression of the full-length *hbox12* mRNA completely reversed this pattern, restraining p38-GFP to the cytoplasm in all blastomeres of early embryos (>80%, n > 250; *Figure 8C*).

To exclude that the ubiquitous p38-GFP nuclear clearance did reflect a visualization artefact, we analyzed the p38 activation in embryos misexpressing *hbox12* in one blastomere at the 4-cell stage, rather than into the zygote. Indeed, in this experimental assay, three-fourths of each embryo that resulted from the uninjected blastomeres provides an internal control. Thus, following injection of the

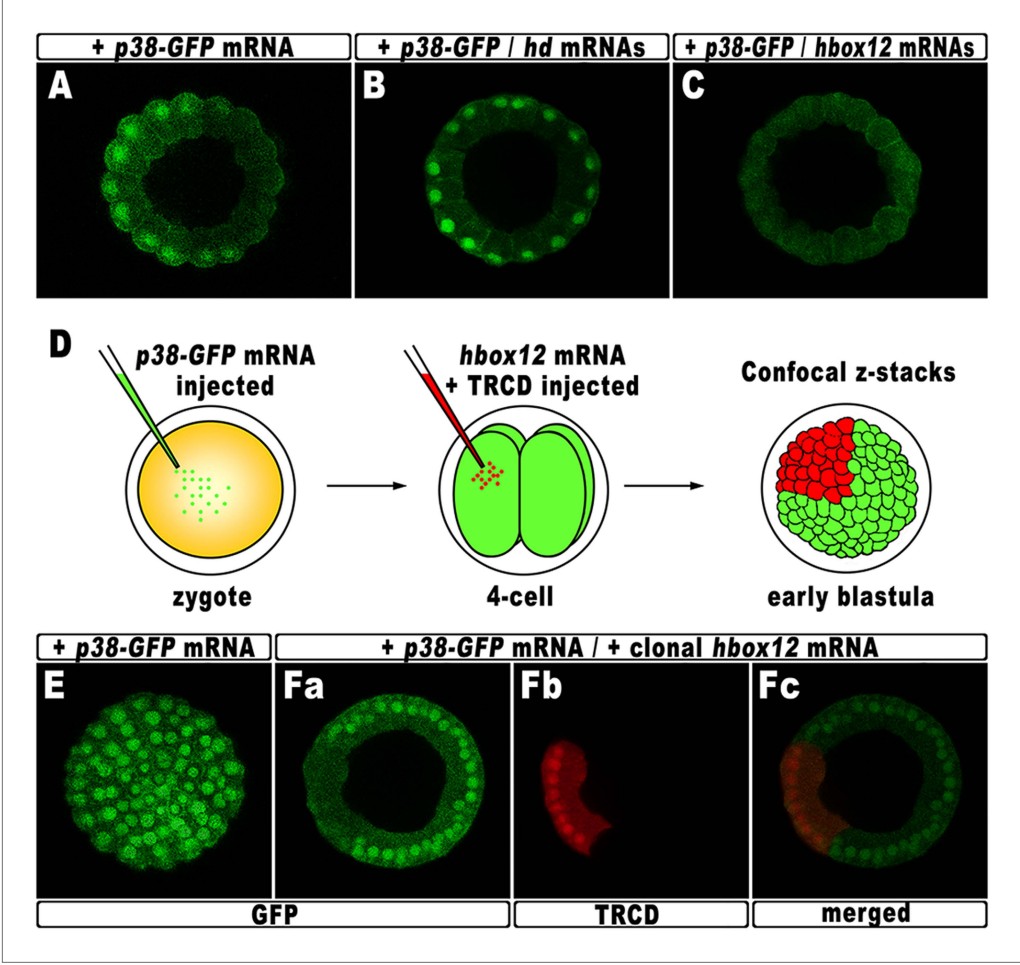

**Figure 8**. Functional correlation between *hbox12* and p38 MAPK. (**A**–**C**) Embryos injected with the *p38-GFP* mRNA either alone (**A**) or in the presence of the *hd* mRNA (**B**) or the *hbox12* mRNA (**C**), respectively, and analyzed by live confocal microscopy at the 60-cell stage. Images are internal projections of 10–15 sections from a confocal z-series. (**D**) At the 4-cell stage, one blastomere of *p38-GFP* mRNA injected embryos was co-injected with the *hbox12* mRNA, and the resulting embryos were examined by live confocal microscopy at the early blastula stage. Representative examples of control *p38-GFP* mRNA injected (**E**) and double-injected (**F**) embryos are shown.

*p38-GFP* mRNA into zygotes, the functional *hbox12* mRNA was injected, along with TRCD, into one blastomere of embryos at the 4-cell stage, and the resulting embryos were scored for GFP localization at the early blastula stage (*Figure 8D*). At this time, active p38 had recovered its ubiquitous distribution, as indicated by the GFP staining present in all nuclei of control embryos (92%, n = 200) injected with the *p38-GFP* mRNA alone (*Figure 8E*). Remarkably, in almost all the double-injected embryos (87%, n > 200), the progeny of the blastomere that received exogenous *hbox12* mRNA firmly detained GFP staining within the cytoplasm (*Figure 8F*), consistent with p38 being a downstream target of Hbox12. Importantly, nuclear TRCD-fluorescence clearly highlights that the *hbox12*-misexpressing cells were perfectly intact and divided synchronously with respect to surrounding cells of the double injected embryos.

It should be emphasized that at the 60-cell stage p38 is normally activated in a sector encompassing about 250° (*Figure 8A*). On the basis of this observation, we reasoned that inhibition of p38 activity through clonal misexpression of *hbox12* could only be attained, at most, in about one third of such a spatial domain. Therefore, it was not surprising to observe that the double-injected embryos developed as normal-looking pluteus larvae (not shown).

To sum up, these results very well correlate with either the ectopic or reduced expression of *nodal* in *hd*-injected and *hbox12* overexpressing embryos, respectively. Most importantly, our findings also strongly support the hypothesis that Hbox12 defines the future dorsal side of the embryo by transient inactivation of p38 activity at a key time point, thereby allowing the asymmetric expression of *nodal* on the opposite side.

## Discussion

### hbox12 and establishing of the DV positional information

During early development of bilaterian embryos symmetry breaking is imposed through establishing of distinct polarities, which are precursors of the larval axes. Although polarization is morphologically not apparent in the zygote, fates of different embryonic regions are patterned on the axial coordinate system by a combination of maternally inherited factors and differential zygotic gene expression (*De Robertis, 2009*).

In sea urchins, the embryonic DV axis formation is intimately linked to a small group of cells that specifically begin to express *nodal* by the early blastula stage and behaves as a DV organizing centre. Production of the Nodal ligand by these cells is pivotal for both defining of their ventral identity and accomplishing the DV patterning program (*Duboc et al., 2004*; *Flowers et al., 2004*; *Yaguchi et al., 2007*). Indeed, Nodal signaling locally induces the production of BMP2/4, a diffusible relay molecule which is translocated on the opposite side of the embryo to specify dorsal cell fates (*Angerer et al., 2000*; *Lapraz et al., 2009*; *Chen et al., 2011*). The DV organizer is also a source of Lefty, which restricts Nodal signaling to the ventral side, and Chordin, which prevents BMP2/4 signaling in the ventral ectoderm (*Bradham et al., 2009*; *Lapraz et al., 2009*). Because polarization requires the concerted action of all these secreted proteins, unveiling the mechanisms that allow competence for spatial positioning of the DV organizer activities is crucial for understanding embryonic patterning.

The data presented here establish that *hbox12* is a key upstream gene in patterning the DV axis of the sea urchin embryo, where it functions to prevent ectopic activation of *nodal* transcription within the prospective dorsal ectoderm. To our knowledge, *hbox12* represents the earliest known zygotic regulatory gene expressed by non-organizer cells and involved in the restriction of the DV organizer field. As a first piece of evidence to support this assertion, we found that overexpression of *hbox12* severely perturbs DV polarity in embryos ubiquitously translating the synthetic full-length *hbox12* mRNA as well as in chimeric specimens bearing the *hbox12* transcript in the animal hemisphere. An almost perfect equivalence of effects is obtained by the expression of HD-En obligate repressor, clearly indicating that Hbox12 acts as a transcriptional repressor in the embryo.

Remarkably, the transcript abundance of *nodal* suffered a dose-dependent attenuation following misexpression of *hbox12* through injection of either *hbox12* or *hd-En* synthetic mRNAs into zygotes. As a consequence, *gsc* and *tbx2/3* genes, which are differentially expressed along the DV axis, downstream to *nodal* signaling, were also down-regulated in both the experimental assays.

The notion of an important role of *hbox12* in limiting the DV organizer function is even further strengthened by the loss-of-function experiments following injection of the *hd* RNA, which revealed that axial specification strayed off the canonical track, eventually culminating in strong ventralization.

In particular, the morphology of the *hd*-injected embryos was steadily spherical during development, never evolving into the characteristic easel-like shape of unperturbed embryos. Their ectoderm did not partition into morphologically distinguishable domains, PMCs were radially distributed around the straight archenteron, forming a mispatterned circular-shaped skeleton, and pigment cells were not produced. As described, all these phenotypic traits have been reported to arise following ectopic expression of *nodal* (***Bradham and McClay, 2006***; ***Duboc et al., 2004***; ***Saudemont et al., 2010***; ***Yaguchi et al., 2010***). Accordingly, molecular analyses revealed a dramatic expansion of the *nodal* expressing domain.

The specificity exerted by HD on both *nodal* and *gsc* gene transcription is well supported by the absence of effect on these marker genes following overexpression of the isolated homeodomain of Otp, indicating individual discrimination of the two homeodomains.

Furthermore, clonal expression of HD imposed by either blastomere transplantation experiments or gene transfer assays unequivocally highlights that Hbox12 action in presumptive dorsal ectoderm cells is necessary to harmoniously pattern the embryo along its DV axis.

This evidence is even better corroborated by the fact that injection of the morpholino oligonucleotide against *nodal* mRNA into a single blastomere at the 4-cell stage could substantially restore the DV polarity of embryos ventralized by the injection of *hd* RNA. Lineage tracing clearly showed that the progeny of the injected blastomere was invariably embedded into the dorsal structures of the rescued larvae. Such a result is fully consistent with the regulative nature of the DV axis specification, historically highlighted since the experiment of Driesch demonstrating the totipotency of each blastomere of the 4-cell stage embryo (***Driesch, 1892***). In fact, cell fates along the DV axis are progressively determined through early development by means of intercellular signaling (***Hurley et al., 1989***; ***Duboc et al., 2004***). Accordingly, *hbox12* transcription is not activated cell-autonomously, being almost undetectable in dissociated embryos (***Di Bernardo et al., 1995***).

Altogether, our findings show that the equilibrium of DV patterning is severely disrupted from the beginning in the absence of the Hbox12 function.

## Hbox12 and the nascent DV gene regulatory network

The events that drive the transcriptional regulation of *nodal* are not completely understood. An interesting line of questioning to pursue in the future would be to evaluate whether Hbox12 directly represses *nodal* transcription in dorsal cells. Intriguingly, several consensus binding sites for homeodomain-containing factors do exist within the promoter sequence of the *nodal* gene (***Range et al., 2007***). On this basis, we cannot exclude the direct association of Hbox12 to the *cis*-regulatory apparatus of *nodal* (***Figure 9***).

On the other hand, several lines of evidence indicate that redox signaling is involved in transcriptional activation of *nodal*. Old and new intriguing experiments suggest that a respiratory gradient in the early sea urchin embryo could bias DV axis orientation (***Pease, 1941***; ***Child, 1948***; ***Czihak, 1963***; ***Coffman and Davidson, 2001***). As an additional clue, the mitochondrial enzyme cytochrome oxidase has higher activity in the presumptive ventral side of the early embryo (***Czihak, 1963***), probably due to a local enrichment in mitochondria (***Coffman et al., 2004***). A possible link between the redox gradient and the initiation of *nodal* transcription is represented by the stress activated kinase p38, which is responsive to reactive-oxygen-species in the sea urchin embryo (***Coffman et al., 2009***) and is required for *nodal* expression (***Bradham and McClay, 2006***). In agreement with this hypothesis, we show that Hbox12 is functionally upstream of p38. This conclusion is firstly supported by the fact that impairing Hbox12 function, by injection of the *hd* RNA, efficiently supplanted the anisotropic inactivation of a GFP-tagged p38 in the prospective dorsal side of the 60-cell stage embryo. By contrast, and in accordance with the previous result, either the ectopic or clonal expression of the *hbox12* mRNA specifically prevents p38-GFP nuclear translocation. These findings strongly suggest that a disrupted distribution of active/inactive p38 realistically accounts for either the ectopic or reduced expression of *nodal* in *hd*-injected and *hbox12* overexpressing embryos, respectively, and concomitant abnormalities in DV patterning.

Taken together, these results show clearly that the negative regulation of *nodal* gene transcription in presumptive dorsal cells is under the control of *hbox12* by means of specific p38 inactivation during early steps of embryogenesis (***Figure 9***). The mechanism by which Hbox12 suppresses p38 function remains an open question. As the functional status of p38 depends upon the balance between specific kinase and phosphatase activities, Hbox12 might switch such a balance in dorsal cells. For instance,

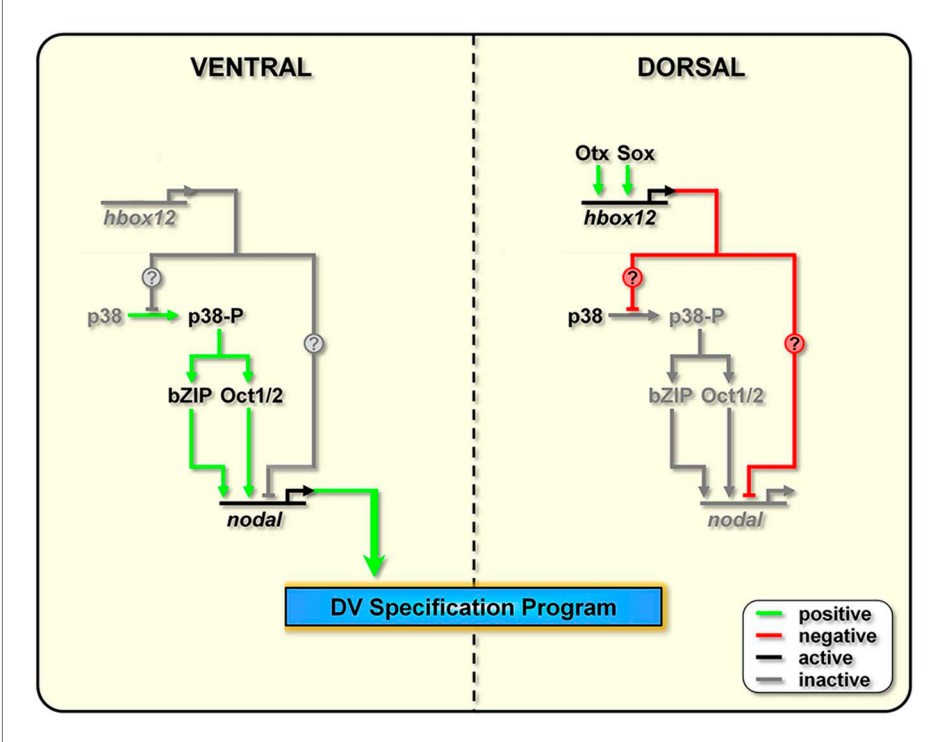

**Figure 9**. Model for establishment of the DV organizing centre in the sea urchin embryo. In the early embryo, *hbox12* transcription is initiated by combinatorial positive inputs from Otx and probably Sox in the future dorsal ectoderm (*Cavalieri et al., 2008*). *hbox12*-dependent suppression of *nodal* gene expression in these cells is mediated by the transient inactivation of p38 and/or probably by direct repression. On the ventral side of the embryo, *hbox12* expression is negatively regulated by unidentified repressors (*Cavalieri et al., 2008*). In these cells, active p38 stimulates *nodal* expression probably through Oct1/2 or other intermediate transcription factors (*Range and Lepage, 2011*), allowing the establishment of the DV organizer and patterning along the secondary axis.

this could be accomplished by suppressing locally the expression of the kinase involved in p38 activation. In a distinct scenario, Hbox12 might work in a regulatory tandem together with a downstream repressor, constituting a so-called double-negative gate which in turn would allow the dorsal-restricted expression of the phosphatase acting on p38. Such an exclusion effect is not uncommon, especially during early embryogenesis, and numerous examples have been described across metazoans (*Oliveri and Davidson, 2007*). Another fascinating possibility will require learning whether p38 directly interacts with the Hbox12 regulator as it does with other homeodomain-containing transcription factors (*Houde et al., 2001*; *Zhou et al., 2013*). Whatever is the mechanism, our results shed new light on the understanding of symmetry breaking events acting upstream of Nodal signaling for the establishment of the DV organizer activity during early embryogenesis of the sea urchin. This makes *hbox12* the earliest regulatory gene integrated in the molecular circuit that initiate DV axis formation in the embryo. The identification and characterization of potential asymmetric input drivers involved in the control of *hbox12* transcription might further improve our understanding of the nature of DV polarization. On this basis, an intriguing issue to be elucidated would be to know whether maternal redox information somehow integrates at the level of the *cis*-regulatory apparatus of the *hbox12* gene. Experiments are currently underway to explore this possibility.

## An evolutionary perspective

Among the paired-like class of homeodomain-containing proteins identified in sea urchins, Pmar1 from *Strongylocentrotus purpuratus* (*Oliveri et al., 2002*), and its ortholog Micro1 from *Hemicentrotus pulcherrimus* (*Kitamura et al., 2002*), appeared to be the most similar to Hbox12. We reasoned that if they were orthologous genes they should show high identity, same *cis*-regulation, as well as spatio-temporal expression pattern, and function.

For instance, the homeodomain of the Otx regulator is 100% identical between *S. purpuratus*, *H. pulcherrimus* and *P. lividus* and fulfils the same function. By contrast, Hbox12 and Pmar1/Micro1 proteins display only an average of 74% identity, with differences found even in the homeodomain.

The spatial expression pattern, and to some extent the timing of activation, of *hbox12* and *pmar1* are profoundly different (*Figure 1*; *Di Bernardo et al., 1995*; *Oliveri et al., 2002*). In addition, while *pmar1* expression is thought to be cell-autonomous in micromeres (*Oliveri et al., 2002*), *hbox12* transcription depends on cell–cell interactions (*Di Bernardo et al., 1995*). Notably, we have demonstrated that the *hbox12* cis-regulatory system is transiently active in the prospective dorsal ectoderm, congruent with the spatiotemporal restriction of the endogenous gene (*Cavalieri et al., 2008*).

Given this premise, a clear cut discordance between the functional outputs of Pmar1 and Hbox12 was expected. Indeed, Pmar1 act as a anti-repressor inhibiting the transcription of the ubiquitous repressor HesC, which negatively regulates the repertoire of early micromere specification genes (*Oliveri et al., 2003*; *Yamazaki et al., 2005*; *Revilla-i-Domingo et al., 2007*). Our results rather suggest an involvement of Hbox12 at the top of the regulatory hierarchy implicated in the polarization of the embryo along the DV axis.

Noteworthy, multiple copies of both *hbox12* and *pmar1/micro1* genes are clustered in *P. lividus* and *S. purpuratus/H. pulcherrimus* genomes respectively, suggesting that numerous rounds of duplication have occurred at these loci during evolution. From this and aforementioned considerations, we suppose that *hbox12* and *pmar1* could be paralogs, rather than orthologs. Considering that the time point for the split between the Strongylocentrotid species and the more recent Parechinid *P. lividus* species is estimated to be about 35–50 Myr (*Smith, 1988*), in such a speculative scenario *hbox12* may have arisen by a *P. lividus* lineage-specific duplication from a common ancestor with *pmar1*.

In this regard, of interest is the sobering clue that the two cited sea urchin species differ in aspects of DV axis determination. For instance, in *S. purpuratus* the DV axis passes through a plane about 45° clockwise from the first cleavage furrow (*Cameron et al., 1989*), indicating that secondary axis specification is initiated between fertilization and first cleavage, which is consistent with the asymmetric distribution of mitochondria within eggs and early embryos of this species (*Coffman et al., 2004*). Classical studies in *P. lividus* embryos instead demonstrate that DV axis is randomly oriented with respect to the first cleavage plane, and that it is established between the fifth and eighth cleavage stage (*Horstadius and Wolsky, 1936*), which broadly corresponds to the peak of *hbox12* expression.

## Materials and methods

### Microinjection, embryo manipulation and imaging

Microinjection was conducted as described (*Cavalieri et al., 2007*, *2009a*). Synthetic mRNAs were resuspended in 30% glycerol and, in selected experiments, TRCD (Molecular Probes, Italy) was added at 5%. For overexpression experiments, capped mRNAs were synthesized from the linearized pCS2 constructs using the mMessage mMachine kit (Ambion, Italy). Approximately 1–2 pl of the purified RNAs were then injected at the following concentrations: *hbox12* and *strim1* out-of-frame, 0.01–0.4 pg/pl; *hd*, *hd-GFP*, and *otp-hd*, 0.4 pg/pl. Mo-*nodal* was instead injected at 0.5 mM along with RDLX. For DNA constructs, approximately 5000 molecules of the linearized *phbox12*-HD-GFP or the control *phbox12*-GFP transgenes were injected per zygote. The *phbox12*-GFP transgene corresponds to the construct initially referred to as 1.45GFP (*Cavalieri et al., 2008*). Such a construct contains a genomic fragment including 1.45 kb of the promoter sequence of *hbox12*, abutting at the 3′ end the ATG start codon fused in frame with the GFP coding sequence (further details are provided in *Cavalieri et al., 2008*). For all experiments, several hundreds of injected embryos were observed and each experiment was repeated at least three times with different batches of eggs.

Recombination of animal and vegetal halves were carried out as described (*Cavalieri et al., 2011*). Briefly, control or injected *P. lividus* embryos at the 16-cell stage were transferred into a modified Kiehart chamber in $Ca^{2+}$-free sea water and manipulated with fine glass needles under a Leica M165FC stereomicroscope equipped with micromanipulators (Narishige, UK). After surgery, the embryos were returned to regular sea water and reared until the desired stage.

Transplanted and non-chimeric injected embryos at the desired stage were harvested, mounted on glass slides and examined under a Leica DM-4500B upright fluorescent microscope. Digital images were captured and processed using Adobe Photoshop CS6.

## Quantitative PCR and whole mount in situ hybridization

Reverse-transcription and qPCR analysis were performed as described (*Cavalieri et al., 2009b*, *2011*, *2013*). Briefly, total RNA from batches of unfertilized eggs and embryos grown at the desired stage was extracted by using the Power SYBR Green Cells-to-CT kit (Ambion, Italy) and reverse transcribed following the manufacturer's recommendations. The resulting cDNA sample was further diluted and the equivalent amount corresponding to one embryo was used as template for qPCR analysis, using the oligonucleotide primers indicated in *Supplementary file 1*. qPCR experiments were performed from two distinct batches and all reactions were run in triplicate on a 7300 Real-Time PCR system (Applied Biosystems, Italy) using SYBR Green detection chemistry. ROX was used as a measure of background fluorescence and, at the end of the amplification reactions, a 'melting-curve analysis' was run to confirm the homogeneity of all amplicons. Calculations from qPCR raw data were performed by the RQ Study software version 1.2.3 (Applied Biosystems, Italy), using the comparative Ct method. Primer efficiencies (i.e., the amplification factors for each cycle) were found to exceed 1.9. In every experiment, a no-template control was included for each primers set. A *cytochrome oxidase* or the *mbf1* mRNA, which are known to be expressed at a constant level during development (*Cavalieri et al., 2008*, *2009b*), were used to normalize all data, in order to account for fluctuations among different preparations.

For developmental expression analysis of *hbox12* and *nodal*, the number of transcripts per embryo at the 60-cell stage was estimated assuming a reference standard number of 1000 copies/embryo of the *z12* mRNA (*Wang et al., 1995*; *Materna et al., 2010*).

Chromogenic whole mount in situ hybridization procedure was performed as described (*Cavalieri et al., 2011*), with Digoxigenin-labeled antisense RNA probes and staged embryos. For the simultaneous detection of *hbox12* and *nodal* transcripts we followed the double two-color WMISH procedure (*Cavalieri et al., 2011*). In this experiment, embryos were co-hybridized with a Digoxigenin-labeled *hbox12* probe along with a Fluorescein-labeled *nodal* probe. Probes were then sequentially revealed using alkaline phosphatase-conjugated antibody with NBT/BCIP (for the dig-probe) or BCIP (for the fluo-probe) as chromogenic substrate.

## Acknowledgements

We express our thanks to Thierry Lepage for the kind gift of the morpholino against *nodal* and the *msp130* cDNA probe; to Christian Gache and Jenifer Croce for sending the *tbx2/3* cDNA probe; to Dave McClay and Cynthia Bradham for generously providing the p38-GFP construct; to Giovanni Morici for initial help with the confocal microscopy.

## Additional information

### Funding

| Funder | Grant reference number | Author |
| --- | --- | --- |
| Università degli Studi di Palermo | STEMBIO award | Vincenzo Cavalieri |
| Università degli Studi di Palermo | ex60% | Vincenzo Cavalieri |
| Assessorato Regionale della Salute, Regione Siciliana | PO FESR 4.1.1.1, RIMEDRI | Vincenzo Cavalieri, Giovanni Spinelli |

The funders had no role in study design, data collection and interpretation, or the decision to submit the work for publication.

### Author contributions

VC, Conception and design, Acquisition of data, Analysis and interpretation of data, Drafting or revising the article, Contributed unpublished essential data or reagents; GS, Conception and design, Analysis and interpretation of data, Drafting or revising the article

## Additional files

**Supplementary file**

• Supplementary file 1. List of gene specific oligonucleotides used in the quantitative RT-PCR.

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
