## [Decision Letter]

The Reviewing editor and the other reviewers discussed their comments before we reached this decision, and the Reviewing editor has assembled the following comments to help you prepare a revised submission.

The reviewers are in agreement that this paper has the potential represent an important contribution to understanding the specification and patterning of the dorsal-ventral (aboral-oral) axis of the sea urchin embryo. This manuscript presents generally compelling experimental evidence that the sea urchin Hbox12 gene encodes a transcriptional repressor whose early expression in the dorsal animal domain of the embryo is required for the proper dorsal-ventral patterning of the embryo by restricting Nodal to the ventral domain. The work contributes to our understanding of early symmetry breaking events in embryonic development and because Nodal has an important role in patterning of other deuterostome embryos this study should be of interest to the broader research community. However there are several important points which need to be addressed in a revised version of the paper before making a final decision on acceptance for publication.

1) The morphological evidence presented strongly supports the conclusions. The analysis of gene expression is somewhat less compelling. It is stated “microinjection of equal amounts of either *hbox12* or *hd-En* mRNA caused a dose-dependent attenuation in the level of nodal transcript at a very similar extent”. However, as mentioned in Figure 2—figure supplement 1 (but not in the Results or Discussion), the reduction in nodal transcript was actually less than significant in the quantitative PCR assay used. The authors reasonably point out that this may be because many of the embryos injected with these mRNAs ultimately did not show loss of dorsal-ventral patterning. This lack of consistency in injected embryos is of some concern and somewhat unusual. The lack of significance (and marginal significance of other qPCR data) should be made clear in the text (and in the legend to Figure 2 which mentions the gray area but not the lack of significance).

2) It is stated ”We conclude that in the double hd-GFP and Mo-nodal injected embryos, clonal knock-down of nodal was able to restore the asymmetrical production of the Nodal ligand, which in turn sufficed to resume the entire DV axis.” While this is a reasonable interpretation, no direct evidence was presented for the abundance or localization of the Nodal (or nodal mRNA) in this experiment, so the statement is conjectural and should be modified.

3) The *hbox12* gene is annotated in the *S. purpuratus* genome as *pmar1* and has been extensively characterized by Oliveri and Davidson. Pmar1 is expressed in micromeres and functions as a critical component of an endomesodermal specification regulatory network. This paper is essentially proposing that there is a radical difference in the function of this gene in two closely related species of urchin. The authors need to make an attempt to provide a reasonable explanation for what many would consider a contradiction. Hence, address a rationale for what appear to be differences in fundamental molecular patterning mechanisms in closely related species.

4) There is a need to see clear data on the localization of the expression of *hbox12*. An underlying assumption of this paper is that expression is restricted to presumptive dorsal ectoderm. The cis-regulatory DNA drives mosaic expression of GFP in dorsal ectoderm, as previously reported, but a conventional in situ hybridization or antibody staining would be reassuring to clearly demonstrate that there is a sound basis for the assumed localization of transcription.

5) General point, it is difficult to come to the conclusions that the authors report without analyzing the distribution of the tissue specific genes that have only been quantified by qPCR; it is essential to have both. Conventional loss of function experiments are not provided data, presumably because there are multiple genes. The RNA overexpression only partly overcomes this problem and the overexpression phenotype is not fully described. For instance, there is a description in the text, and a figure (Figure 2), but the only substantiation of changes in tissue distribution is a low quality in situ RNA hybridization for a mesenchyme gene. If the contention is that there is a change in the distribution of ectodermal domains, this really should be substantiated with tissue specific gene or antibody localizations with specific markers.

6) Specific expression analysis this would help other parts of the paper as well, for instance in the Results section, the defects are not clearly dorsal-ventral polarity and tissue specific markers should be used to establish this. Similarly the authors assert a dorsal-ventral defect without demonstrating alterations in the distribution of tissue markers.

7) A minor point, a curious outcome of the experiments is the effects on the pigment cell lineage. As this lineage arises from mesoderm, the effect is not fully explained. A clear demonstration that the early commitment of macromere derived cells is the proximal effect by using tissue specific gene probes; the standard in the field would be valuable. The loss of pigment and the suppression of GCM could come about in several ways: one cannot assume it is at the level of specification.

---

## [Author Response]

*1) The morphological evidence presented strongly supports the conclusions. The analysis of gene expression is somewhat less compelling. It is stated “microinjection of equal amounts of either* hbox12 *or* hd-En *mRNA caused a dose-dependent attenuation in the level of nodal transcript at a very similar extent”. However, as mentioned in*
Figure 2—figure supplement 1
*(but not in the Results or Discussion), the reduction in nodal transcript was actually less than significant in the quantitative PCR assay used. The authors reasonably point out that this may be because many of the embryos injected with these mRNAs ultimately did not show loss of dorsal-ventral patterning. This lack of consistency in injected embryos is of some concern and somewhat unusual. The lack of significance (and marginal significance of other qPCR data) should be made clear in the text (and in the legend to*
Figure 2
*which mentions the gray area but not the lack of significance)*.

As far as qPCR data is concerned, we would like to emphasize that the 3-fold significance threshold does not correspond to a technical limit of our experimental conditions. Rather, it is the criterion proposed by Davidson (reviewed in Davidson et al., 2002, Science 295, 1669-78) and widely adopted by the rest of the sea urchin community. However, less than 3-fold difference could be significant when strengthened by WMISH results, and this is exactly our case.

Since only about half of the *hbox12*-injected embryos developed as aberrant larvae (virtually indistinguishable from the unaffected ones at early stages), we predicted that the qPCR data on *nodal*, *gsc* and *tbx2/3* expression would refer to an heterogeneous population of embryos (affected and not affected).

Thus, we performed WMISH on control- and *hbox12*- injected embryos at the mesenchyme blastula stage, in which DV polarity is not yet morphologically noticeable. In close agreement with our hypothesis, a significant fraction (∼47%) of *hbox12*-injected embryos showed reduced expression of the mentioned genes. Most importantly, such a fraction was reasonably congruent with that of embryos that ultimately displayed unambiguous defects in DV polarization at later stages.

These results were included in the current version of the manuscript (in the Results section and Figure 2—figure supplement 2), and a detailed explanation concerning the suspected lack of significance of qPCR data was given both in the main text and Figure 2 legend, as requested.

*2) It is stated “We conclude that in the double hd-GFP and Mo-nodal injected embryos, clonal knock-down of nodal was able to restore the asymmetrical production of the Nodal ligand, which in turn sufficed to resume the entire DV axis.” While this is a reasonable interpretation, no direct evidence was presented for the abundance or localization of the Nodal (or nodal mRNA) in this experiment, so the statement is conjectural and should be modified*.

As suggested, the statement has been modified.

*3) The* hbox12 *gene is annotated in the* S. purpuratus *genome as* pmar1 *and has been extensively characterized by Oliveri and Davidson. Pmar1 is expressed in micromeres and functions as a critical component of an endomesodermal specification regulatory network. This paper is essentially proposing that there is a radical difference in the function of this gene in two closely related species of urchin. The authors need to make an attempt to provide a reasonable explanation for what many would consider a contradiction. Hence, address a rationale for what appear to be differences in fundamental molecular patterning mechanisms in closely related species.*

In the former version of the submitted manuscript we did not mention neither *pmar1* nor that *hbox12* and *pmar1* are orthologous genes. In fact, we believe that *hbox12* and *pmar1* are indeed distinct genes playing distinct roles in two evolutionary distant species of sea urchins. To meet the Referee’s request, we have included a paragraph in the Discussion section to explain this point.

*4) There is a need to see clear data on the localization of the expression of* hbox12*. An underlying assumption of this paper is that expression is restricted to presumptive dorsal ectoderm. The cis-regulatory DNA drives mosaic expression of GFP in dorsal ectoderm, as previously reported, but a conventional in situ hybridization or antibody staining would be reassuring to clearly demonstrate that there is a sound basis for the assumed localization of transcription*.

The dorsal-specific expression of *hbox12* was not arbitrarily assumed, but rather derived from previous results based on in situ hybridization ([28], *Proc. Natl. Acad. Sci. U.S.A.* 92, 8180-8184.) and *cis*-regulatory analysis ([12], *Dev. Biol.* 321, 455-469).

To definitively confirm the dorsal localization of *hbox12* transcripts, we performed double-staining WMISH with probes for *hbox12* and *nodal* (which marks the ventral side of the developing embryo). As clearly shown in Figure 1, the spatial domains of expression of the two genes occupy opposite sectors of the embryo at the early blastula stage.

*5) General point, it is difficult to come to the conclusions that the authors report without analyzing the distribution of the tissue specific genes that have only been quantified by qPCR; it is essential to have both. Conventional loss of function experiments are not provided data, presumably because there are multiple genes. The RNA overexpression only partly overcomes this problem and the overexpression phenotype is not fully described. For instance, there is a description in the text, and a figure (*Figure 2*), but the only substantiation of changes in tissue distribution is a low quality in situ RNA hybridization for a mesenchyme gene. If the contention is that there is a change in the distribution of ectodermal domains, this really should be substantiated with tissue specific gene or antibody localizations with specific markers*.

As suggested, we answered to this point by performing WMISH with specific probes for ectodermal domains and implemented the current version of the manuscript accordingly (see also point 1 of this letter for a more detailed answer).

*6) Specific expression analysis this would help other parts of the paper as well, for instance in the Results section, the defects are not clearly dorsal-ventral polarity and tissue specific markers should be used to establish this. Similarly the authors assert a dorsal-ventral defect without demonstrating alterations in the distribution of tissue markers*.

This point refers to the results derived from the transplanted chimeric embryos. We are well aware that the suggested additional analysis of tissue markers in chimeras would improve our paper but, unfortunately, we face two types of difficulties that make these experiments quite unfeasible, at least in this period of the year. This is because currently we are not in the spawning season. In Sicily, the *P. lividus* reproductive cycle starts with the New Year but eggs are not suitable for microinjection and/or micromanipulation before the end of March. This would delay the publication of our paper quite a lot.

In addition, there is also a severe adverse technical limitation related to the fact that several tens of micro-injection/manipulation sessions would be required in order to obtain relevant numbers of chimeras to be processed for in situ hybridization with various probes. This would introduce a high extent of variability due to mixing different batches of embryos (in our hands the number of chimeric embryos is about ten for each experiment), that might hinder the interpretation of the results.

On the other hand, we would like to underlie that, in our opinion, the morphological evidence presented fit well with all the other results and support our conclusions.

*7) A minor point, a curious outcome of the experiments is the effects on the pigment cell lineage. As this lineage arises from mesoderm, the effect is not fully explained. A clear demonstration that the early commitment of macromere derived cells is the proximal effect by using tissue specific gene probes; the standard in the field would be valuable. The loss of pigment and the suppression of GCM could come about in several ways: one cannot assume it is at the level of specification*.

We fully agree with the concept that loss of pigment and *gcm* suppression in HD-expressing embryos may not necessarily reflect a failure in the specification of secondary mesenchyme cells. However, this appear to us the most plausible explanation essentially because *hd*-injected embryos are subjected to overexpression of *nodal*, and it is commonly accepted that Nodal signaling antagonizes the specification of pigment cells in unperturbed embryos. In agreement to this notion, it has been demonstrated by other authors that ubiquitous expression of *nodal* mRNA provokes loss of *gcm* transcription in the resulting albino embryos ([31], *Development* 137, 223-235). A circumstantiated explanation has been included in the Results section.